# Insights into the RNA Virome of the Corn Leafhopper *Dalbulus maidis*, a Major Emergent Threat of Maize in Latin America

**DOI:** 10.3390/v16101583

**Published:** 2024-10-09

**Authors:** Humberto Debat, Esteban Simon Farrher, Nicolas Bejerman

**Affiliations:** 1Instituto de Patología Vegetal—Centro de Investigaciones Agropecuarias—Instituto Nacional de Tecnología Agropecuaria (IPAVE-CIAP-INTA), Camino 60 Cuadras Km 5.5, Córdoba X5020ICA, Argentina; 2Unidad de Fitopatología y Modelización Agrícola—Consejo Nacional de Investigaciones Científicas y Técnicas (UFYMA-CONICET), Camino 60 Cuadras Km 5.5, Córdoba X5020ICA, Argentina; 3Facultad de Ciencias Químicas, Universidad Nacional de Córdoba, Córdoba 5000, Argentina; esteban.farrher@mi.unc.edu.ar

**Keywords:** RNA virome, *Dalbulus maidis*, maize leafhopper, data mining

## Abstract

The maize leafhopper (*Dalbulus maidis*) is a significant threat to maize crops in tropical and subtropical regions, causing extensive economic losses. While its ecological interactions and control strategies are well studied, its associated viral diversity remains largely unexplored. Here, we employ high-throughput sequencing data mining to comprehensively characterize the *D. maidis* RNA virome, revealing novel and diverse RNA viruses. We characterized six new viral members belonging to distinct families, with evolutionary cues of beny-like viruses (*Benyviridae*), bunya-like viruses (*Bunyaviridae*) iflaviruses (*Iflaviridae*), orthomyxo-like viruses (*Orthomyxoviridae*), and rhabdoviruses (*Rhabdoviridae*). Phylogenetic analysis of the iflaviruses places them within the genus *Iflavirus* in affinity with other leafhopper-associated iflaviruses. The five-segmented and highly divergent orthomyxo-like virus showed a relationship with other insect associated orthomyxo-like viruses. The rhabdo virus is related to a leafhopper-associated rhabdo-like virus. Furthermore, the beny-like virus belonged to a cluster of insect-associated beny-like viruses, while the bi-segmented bunya-like virus was related with other bi-segmented insect-associated bunya-like viruses. These results highlight the existence of a complex virome linked to *D. maidis* and paves the way for future studies investigating the ecological roles, evolutionary dynamics, and potential biocontrol applications of these viruses on the *D. maidis*—maize pathosystem.

## 1. Introduction

The maize leafhopper *Dalbulus maidis* (DeLong & Wolcott) (Hemiptera: Auchenorrhyncha) is a significant threat to maize crops causing extensive economic losses [1]. This insect pest is widely distributed in the Americas, from the southern United States to northern Argentina [2]. In addition to the direct damage caused by its feeding, this insect also serves as the vector for the mollicute *Spiroplasma kunkelii* Whitcomb, the causal agent of Corn Stunt disease, and the maize bushy stunt (MBS) phytoplasma [1]. *D. maidis* is also the vector of two viruses: maize rayado fino virus (MRFV) [1], and maize striate mosaic virus (MSV) [3]. While its ecological interactions [4,5] and control strategies, such as germplasm resistance, insecticide sprays, and some cultural practices are well studied [5,6], its associated viral diversity remains largely unexplored.

Viruses are the most abundant microbes on the planet [7,8,9] and RNA viruses dominate the eukaryotic virome [10,11]. Hundreds of viruses have been identified in recent years in arthropods [10,11,12,13,14,15], illustrating that invertebrates usually have a greater and more diverse virome than vertebrates [16]. Nevertheless, over the last years, there has been a rapid increase in vertebrate-associated virus entries in the NCBI-GenBank, while low numbers of invertebrate-associated virus have been identified and uploaded into public repositories [16]. The availability of a great amount of RNA-seq data (metatrancriptomics) in public databases has revolutionized the field of virus discovery, fostering the identification of novel viruses even without a previous knowledge of their genome sequence [7,17,18]. The comprehensive analysis of the RNA virosphere led to the identification of a great number of novel viruses, thus shedding light not only on the huge diversity of RNA viruses in arthropods, but also on their evolutionary histories as well as on their genomic organization complexity [11,12,15,16,19,20]. Furthermore, these advances in the knowledge of the virosphere are leading to opportunities and challenges in virus discovery [21] and to develop a new, genome-based virus taxonomy [22,23,24,25,26].

This paper reports the analysis of 23 publicly available *D. maidis* transcriptome datasets from USA, Brazil, and Argentina, that resulted in the discovery of six novel RNA viruses associated with this insect. To our knowledge, this is the first report of insect viruses associated with this economically important insect, revealing that the *D. maidis* RNA virome is rich and diverse.

## 2. Material and Methods

### 2.1. Exploration of the D. maidis Virome from Public RNA-seq Datasets, Sequence Assembly, and Virus Identification

We analyzed all *D. maidis* Sequence Read Archives (SRA) deposited in the NCBI (https://www.ncbi.nlm.nih.gov/sra, (accessed on 22 April 2024)). The raw nucleotide sequence reads from each SRA experiment were downloaded from their associated NCBI BioProjects (Appendix A). The datasets were pre-processed by trimming and filtering with the Trimmomatic v0.40 tool as implemented in http://www.usadellab.org/cms/?page=trimmomatic (accessed on 23 April 2024) with standard parameters. The resulting reads were assembled de novo with rnaSPAdes using standard parameters on the Galaxy server (https://usegalaxy.org/). The transcripts obtained from de novo transcriptome assembly were subjected to bulk local BLASTX searches (E-value < 1 × 10^−5^) against a curated database of the publicly available reference viral proteins. The resulting viral sequence hits of each dataset were explored in detail. Tentative virus-like contigs were curated (extended and/or confirmed) by iterative mapping of each SRA library’s filtered reads. This strategy was used to extract a subset of reads related to the query contig, used the retrieved reads from each mapping to extend the contig and then repeat the process iteratively using as query the extended sequence [27]. The extended and polished transcripts were reassembled using the Geneious v8.1.9 (Biomatters Ltd., Auckland, New Zealand) alignment tool with high sensitivity parameters.

### 2.2. Bioinformatics Tools and Analyses

#### 2.2.1. Sequence Analyses

Open reading frames (ORFs) were predicted with ORFfinder (minimal ORF length 120 nt, genetic code 1, https://www.ncbi.nlm.nih.gov/orffinder/, (accessed on 7 May 2024)), functional domains, and the architecture of translated gene products were determined using InterPro (https://www.ebi.ac.uk/interpro/search/sequence-search (accessed on 8 May 2024)) and the NCBI Conserved domain database—CDD v3.20 (https://www.ncbi.nlm.nih.gov/Structure/cdd/wrpsb.cgi (accessed on 8 May 2024)) with E-value = 0.1. Further, HHPred and HHBlits as implemented in https://toolkit.tuebingen.mpg.de/#/tools/ (accessed on 8 May 2024) were used to complement annotation of divergent predicted proteins by hidden Markov models. Transmembrane domains were predicted using the TMHMM version 2.0 tool (http://www.cbs.dtu.dk/services/TMHMM/, (accessed on 8 May 2024)) and signal peptides were predicted using the SignalP version 6.0 tool (https://services.healthtech.dtu.dk/services/SignalP-6.0/, (accessed on 8 May 2024)), while nuclear localization signals (NLS) were predicted using cNLS Mapper tool (https://nls-mapper.iab.keio.ac.jp/cgi-bin/NLS_Mapper_form.cgi, (accessed on 8 May 2024)). The presence of gene junction sequences flanking ORFs was also included as a criterion to determine the potential coding sequences. The predicted proteins were then subjected to NCBI-BLASTP searches against the non-redundant protein sequences (nr) database.

#### 2.2.2. Phylogenetic Analysis

Phylogenetic analysis based on the RdRp proteins, was carried out using MAFFT 7.505 with multiple aa sequence alignments using FFT-NS-i as the best-fit model. The aligned aa sequences were used as the input in Mega11 software [28] to generate phylogenetic trees by the maximum-likelihood method using the best-fit substitution model. Local support values were computed using bootstraps with 1000 replicates.

## 3. Results and Discussion

### 3.1. RNA Viruses Associated with D. maidis

Insects are the most abundant group of animals on earth [29]. These animals have a rich and diverse virome which is being unraveled in the later years using high-throughput sequencing [10,11,14,15,30]; however, only a small portion of the insect viromes diversity has been described so far [31]. Several viruses were identified in some leafhoppers, mainly iflaviruses [14]. In addition, a rhabdovirus [32], an orthomyxo-like virus [12], a chuvirus [33], a reovirus [34], a nido-like virus [35], and nine RNA viruses, belonging to seven viral clades (Picorna-Calici, Permutotetra, Bunya-Arena, Reo, Partiti-Picobirna, Luteo-Sobemo and Toti-Chryso) [36], were also identified, which illustrates how rich and diverse the virome associated with leafhoppers could be. There are more than 20,000 described species of leafhoppers (*Cicadellidae*) which illustrate that the handful of viruses linked to this invertebrate is scarce, poorly misrepresents the diversity of these invertebrates, and even important agronomic pests as the maize leafhopper virome are still unexplored. Thus, here we analyzed transcriptome datasets of maize leafhopper collected in Argentina, USA, and Brazil. The de novo assembling of those 23 libraries followed by similar analyses resulted in the robust identification of six novel highly divergent RNA viruses, including a beny-like virus, a bunya-like virus, two iflaviruses, a orthomyxo-like virus, and a rhabdovirus, which were tentatively named Chicharrita del maiz beny-like virus (ChMBLV), Chicharrita del maiz bunya-like virus (ChMBV), Chicharrita del maiz iflavirus 1 (ChMIfV1), Chicharrita del maiz iflavirus 2 (ChMIfV2), Chicharrita del maiz orthomixo-like virus (ChMOMLV) and Chicharrita del maiz rhabdoirus (ChMRV), respectively. This first glimpse of the virus repertoire reveals how rich and diverse the virome associated with the maize leafhopper is. We assembled the full-length coding region sequence for all viruses, but the rhabdovirus, for which its nearly complete coding region was able to be assembled. Revealing the virus diversity in this organism is crucial to further exploit the identified viruses as specific and targeted tools for the biological control of insect populations.

### 3.2. Molecular and Phylogenetic Characterization of the Newly Identified Maize Leafhopper Beny-Like Viruses

The *Benyviridae* is a family with members that have positive-sense, single stranded RNA viruses with only one genus, the *Benyvirus*, recognized by the ICTV [37]. Benyviruses are plant-infecting viruses with a multipartite genome [37]. Nevertheless, metagenomic studies as well as the metatrancriptomic analysis of public data deposited in NCBI resulted in the discovery of many benyvirus-related sequences, that resemble to newly unassigned *Benyviridae* members, that likely infect a wide range of non-plant hosts, such as insects and fungi [38,39,40,41]. Moreover, the insect-associated beny-like virus genomes are monopartite which is contrasting to the multipartite genomes displayed by the plant-associated counterparts. Here, a novel beny-like virus was identified and named as Chicharrita del maiz beny-like virus (ChMBLV), which was only detected in one transcriptome dataset of maize leafhopper, that was collected in the USA (Appendix A). The positive-sense, single stranded RNA ChMBLV sequence (GenBank accession number BK068250) is 6887 nucleotides (nt) long and has two ORFs encoding a non-structural polyprotein of 2003 amino acids (aa) and a putative structural protein of 224 (Figure 1A) (Table 1). This genomic organization is similar to that reported for the beetle-associated viruses Diabrotica undecimpunctata virus 2 (DuV2) [42] and Guiyang benyvirus 1 (GuBV1) (Feng, unpublished). ChMBLV non-structural polyprotein showed the highest BlastP similarity with that one encoded by DuV2, with 46.61% identity (Table 1); therefore, based on genetic distance, ChMBLV is a new virus distantly related to other NCBI database entries of insect viruses. Three conserved domains, Alphavirus-like methyltransferase (MT) domain, the Viral_helicase1, and the ps-ssRNAv_RdRp-like, were identified in the polyprotein (Figure 1A) at aa positions 41–386, 617–848 and 1673–1857, respectively. Similar conserved domains were found in those non-structural polyproteins encoded by related beny-like viruses [38,39,41,42]. Unlike its plant-associated counterparts [37,41], insect-associated beny-like virus replicase, like those encoded by ChMBLV and its related viruses [38,39,40,42], do not have any papain-like protease domain in their sequences. ChMBLV structural protein showed the highest BlastP similarity with the one encoded by DuV2, with 29.75% identity (Table 1). This protein encoded by the ChMBLV ORF2 is likely a coat protein, since the TMV_coat super family conserved domain was identified (Figure 1A) at aa position 108–180. A similar conserved domain was also identified in the ORF2-encoded proteins of DuV2 [42], and GuBV1 (Feng, unpublished).

Phylogenetic analysis of the non-structural polyprotein amino acid sequence placed ChMBLV in a clade containing several insect-associated beny-like viruses, where ChMBlV was clustered together with the beetle Lepidiota negatoria beny-like virus within the insect clade of beny-like viruses, which is separated from the clades composed of fungi-associated beny-like viruses and plant-associated ones (Figure 1B). Thus, we propose to create a new genus, to be named as “*Insebenyvirus*” within the family *Benyviridae*, to accommodate the insect-associated beny-like viruses. Based on the phylogenetic insights and the observed genetic distance of the newly identified viruses, we tentatively propose an aa sequence identity of 80% in the CP as threshold for species demarcation in this newly proposed genus.

To our knowledge, ChMBLV is the first reported beny-like virus associated with leafhoppers.

### 3.3. Molecular and Phylogenetic Characterization of the Newly Identified Maize Leafhopper Bunya-Like Virus

*Bunyavirales* is a viral order including several families of negative-sense single-stranded RNA (-ssRNA) viruses with a multi segmented genome [43]. Bunyavirids are highly diverse and have a wide host range and a considerable genome complexity. Many proposed members of this family have been discovered in several insect species [12]. Here, a novel bipartite bunya-like virus was identified and named as Chicharrita del maiz bunyan-like virus (ChMBV), which was detected in the three transcriptome datasets of maize leafhopper from Brazil (Appendix A). The full-length ChMBV coding sequence (GenBank accession numbers BK068254 and BK068255) comprised two segments (L and S) of single-stranded, negative-sense RNA of 6651 nt and 1281 nt, respectively. Segment L encodes a putative RdRp protein of 2166 aa, and segment S encodes a putative nucleoprotein (NP) of 327 aa (Figure 2A) (Table 1).

As expected, the structural NP encoding segment was the most abundant one in the three libraries from Brazil, whereas the segment encoding the RdRp was the one that accumulated less on those datasets where ChMBV was identified (Appendix A). ChMBV presents low homology to other bunyavirids (Table 1). The RdRp protein shows the highest BlastP similarity with the leafhopper-associated Hubei insect virus 1 (HbIV1) RdRp, with 49.84% identity (Table 1), and a Bunya_RdRp super family conserved domain was identified in its sequence at aa positions 603–1303. The NP protein shows the highest BlastP similarity with the HbIV1 NP, with 38.01% identity (Table 1), and a Tenui_N super family conserved domain has been identified in its sequence at aa positions 40–191.

The phylogenetic tree based on the ChMBV RdRp-encoded protein showed that this virus is evolutionarily related to the bipartite leafhopper-associated bunya-like viruses. ChMBV clustered with HbIV1, Hemipteran phenui-related virus OKIAV285 and the rice green leafhopper-associated virus Hangzhou nephotettix cincticeps phenuivirus 1, and this cluster is linked with the planthopper-associated Sanya nilaparvata lugens phenuivirus 1 (Figure 2B). Bunyavirids have a different number of genome segments; however, those viruses phylogenetically related with ChMBV are all bisegmented RNA viruses [11]. Intriguingly, the segment coding for the glycoprotein was not identified in ChMBV and in those related viruses. Thus, these leafhopper-associated bunya-like viruses appear to be a new cluster of bisegmented RNA viruses that could be classified into a new genus within the *Bunyavirales* order. Based on the phylogenetic insights and the observed genetic distance of the newly identified viruses, we tentatively propose an aa sequence identity of 80% in the L protein as threshold for species demarcation in this newly proposed genus.

### 3.4. Molecular and Phylogenetic Characterization of the Newly Identified Maize Leafhopper Iflaviruses

*Iflaviridae* is a family composed of only one genus, the *Iflavirus*, of positive-sense single-stranded RNA (-ssRNA) viruses with a monopartite genome, that contains an ORF encoding a single polyprotein of approximately 3000 aa. All reported iflaviruses are associated with arthropod hosts, mainly insects [44,45], and are among the most common viruses identified in different species of arthropods [46]. Here, two iflaviruses were identified and named as Chicharrita del maiz iflavirus virus 1 (ChMIfV1) and Chicharrita del maiz iflavirus 2 (ChMIfV2). ChMIfV1 was detected in five transcriptome datasets of the maize leafhopper (one from Argentina, another from USA and three from Brazil), while ChMIfV2 was detected in 18 transcriptome datasets, all from USA (Appendix A). The iflaviruses were the most prevalent and abundant among the six viruses identified in this study (Appendix A).

The positive-sense single-stranded RNA ChMIfV1 genome (GenBank accession number BK068251) is 11,104 nt in length, while the ChMIfV2 genome (GenBank accession number BK068252) is 10,318 nt in length (Figure 3A) (Table 1) and are both A/U rich, similar to other iflaviruses [47]. Both viruses have a single ORF which encodes a polyprotein of 3306 aa and 3102 aa, respectively (Figure 3A) (Table 1).

ChMIfV1 and ChMuIfV2 polyproteins share only a 26.3% aa identity between them, and ChMIfV1 polyprotein showed the highest BlastP similarity with that one encoded by the rice green leafhopper-associated virus Nephotettix cincticeps positive-stranded RNA virus 1 (NcPSRV1), with 64.03% identity (Table 1); while ChMIfV2 polyprotein showed the highest BlastP similarity with that one encoded by the American grapevine leafhopper-associated virus Scaphoideus titanus iflavirus 1 (StIfV1) with 53.04% identity (Table 1). The prediction of the conserved domains present in both ChMIfV1 and ChMuIfV2 polyproteins showed that both viruses have the typical iflavirus gene organization. The structural domains are located at the N-terminal region of the polyprotein, while the non-structural domains are located downstream in the C-terminal region [44]. Two rhinovirus-like capsid domains (Rhv-like, aa 338–518 and aa 597–785 in ChMIfV1, aa 327–508 and aa 602–786 in ChMIfV2) and a cricket paralysis virus–like capsid domain (CRPV_capsid super family, aa 997–1148 im ChMIfV1 and aa 1057–1209 in ChMIfV2) were identified in the N-terminal region of the ChMIfV1 and ChMIfV2 polyproteins, while an helicase (RNA_Helicase, aa 1550–1656 in ChMIfV1 and aa 1552–1658 in ChMIfV2), a Picornavirales 3C/3C-like protease (3C-Pro, aa 2529–2741 in ChMIfV1 and aa 2301–2530 in ChMIfV2), and RNA-directed RNA polymerase (RdRp, aa 2952–3256 in ChMIfV1 and aa 2738–3055 in ChMIfV2) conserved domains were identified in the C-terminal region of the ChMIfV1 and ChMuIfV2 polyproteins (Figure 3A).

Phylogenetic analysis based on the RdRp conserved region shows that ChMIfV1 and ChMIfV2 clustered within the *Iflaviridae* members with other leafhopper-associated iflaviruses; where both viruses were placed in different clades as predicted by the low identity observed between them (Figure 3B). ChMIfV1 clustered with NcPSRV1 and the rice pest “zig-zag leafhopper”-associated virus Hangzhou recilia dorsalis iflavirus 1 (Figure 3B); while ChMIfV2 clustered with the American grapevine leafhopper StIfV1 and StIfV2, Hangzhou recilia dorsalis iflavirus 2, and the rice green leafhopper-associated virus Congyang nephotettix cincticeps iflavirus 1 (Figure 3B). All in all, the branching of both viruses within the group of Iflavirus members indicates that both viruses should be included in the family *Iflaviridae* (Figure 3B). According to the demarcation criteria set for ilfaviruses [44], ChMIfV1 and ChMIfV2 should be classified as two novel species named *Iflavirus alphadalbuli* and *Iflavirus betadalbuli*.

Iflavirus infections are usually asymptomatic; however, some ifllaviruses have been reported to be harmful to the host insects leading to morphological deformities, behavioral change, and death [48,49,50]. Moreover, recently the armyworm Spodoptera exigua iflavirus 1 (SeIV1) was shown to enhance the susceptibility of the armyworm to the disease caused by Spodoptera exigua multiple nucleopolyhedrovirus (SeMNPV) [51], as well as its susceptibility to Bacillus thuringiensis-based insecticide formulations and the parasitism by the wasp Hyposoter didymator [52]. Moreover, iflaviruses have been proposed for VIGS to regulate the expression of insect metabolic genes [53]. Thus, it would be critical to further explore if the iflaviruses identified in this study are pathogenic to their insect host for their potential deployment as tools for biocontrol agents.

### 3.5. Molecular and Phylogenetic Characterization of the Newly Identified Maize Leafhopper Orthomyxo-Like Virus

*Orthomyxoviridae* is a family of negative-sense single-stranded RNA (-ssRNA) viruses with a multi segmented genome [54]. Members belonging to the *Orthomyxoviridae* family are highly diverse, with a wide host range and complex genomes. Many members of this family have been discovered in various insect species [12]. Here, a novel orthomyxo-like virus was identified and named as Chicharrita del maiz orthomyxo-like virus (ChMOMLV), which was detected in three transcriptome datasets of maize leafhopper, one from USA and two from Brazil. The library from USA showed the highest abundance of virus reads, and among segments, as expected, the ones encoding for the structural nucleoprotein (NP) and hemagglutinin protein (HA) showed the highest RNA titers in all ChMOMLV-positive libraries (Appendix A). The ChMOMLV genome sequence (GenBank accession numbers BK068245-BK068250) comprised five segments of single-stranded, negative-sense RNAs of 2557 nt, 2460 nt, 2348 nt, 1817 nt, and 1681 nt (Figure 4A). RNA1 and RNA2 encode polymerase PB1 and PB2 proteins of 794 and 777 aa, respectively, and RNA3 encodes a putative polymerase PA of 739 aa. RNA4 encodes a putative NP of 572 aa, while RNA5 encodes a putative HA of 503 aa (Figure 4A) (Table 1). ChMOMLV presents low homology to other orthomixo-like virus and members belonging to the *Orthomyxoviridae* family (Table 1). PB1 protein shows the highest BlastP similarity with the brown marmorated stink bug associated virus Halyomorpha halys orthomyxo-like virus 1 (HhOlV1) PB1, with 49.41% identity, and a Flu_PB1 super family conserved domain has been identified in its sequence at aa positions 41–747. PB2 protein shows the highest BlastP similarity with HhOlV1-PB2, with 26.32% identity and a Flu_PB2 super family conserved domain has been identified in its sequence at aa positions 15–409. The PA protein shows the highest BlastP similarity with the Bat faecal-associated orthomyxo-like virus 1 PA, with 34.59% identity and a Flu_PA super family conserved domain was detected in its sequence at aa positions 126–731. The putative NP protein shows the highest BlastP similarity with the Hemipteran orthomyxo-related virus OKIAV183 NP, with a 29.87% identity with a Flu_NP super family conserved domain in its sequence at aa positions 340–553. The HA protein shows the highest BlastP similarity with the parasitoid wasp-associated virus Cotesiavirus orthomyxo HA, with 24.56% identity and an envelope glycoprotein conserved domain at aa positions 295–495.

The phylogenetic tree based on the ChMOMLV PB1-encoded protein, showed that this virus is clustered with the insect-associated orthomyxo-like viruses HhOlV1 and Hemipteran orthomyxo-related virus OKIAV191 and this cluster is related to a clade formed by putative quaranjavirus (Figure 4B). Orthomixovirids have different number of genome segments; however, those viruses phylogenetically related to ChMOMLV also have five RNA segments [12,55]. Viruses belonging to the genus *Quaranjavirus* are known as argasid tick-borne viruses and have genomes with five or six RNA segments [56]. Many quaranji-like viruses were recently described in several insects; therefore, the diversity of quaranjaviruses might be greatly underestimated in arthropods other than the previously described hematophagous insects [39]. Thus, future studies should assess whether this group of unclassified quaranja-like viruses including ChMOMLV could be classified within the genus *Quaranjavirus*, or if would be more appropriate to encompass these fifth segmented viruses in a novel genus within the family *Orthomyxoviridae*. Based on the phylogenetic insights and the observed genetic distance of the newly identified viruses, we tentatively propose an aa sequence identity of 80% in the PB1 protein as a threshold for species demarcation in the newly proposed genus.

To our knowledge, ChMOMLV is the first reported orthomyxo-like virus associated with leafhoppers.

### 3.6. Molecular and Phylogenetic Characterization of the Newly Identified Maize Leafhopper Rhabdovirus

*Rhabdoviridae* is a family of negative-sense single-stranded RNA (-ssRNA) viruses with a monopartite genome as a hallmark of almost all rhabdoviruses. However, plant-associated rhabdoviruses with bi-segmented and tri-segmented genomes have been described [27,57]. Members belonging to the *Rhabdoviridae* family are highly diverse and have a wide host range and a great genome complexity and many members of this family have been discovered in many insect species, some of which may serve as unique hosts or may act as biological vectors for transmission to other animals or plants [12,57,58]. Here, a novel rhabdo-like virus was identified and named as Chicharrita del maiz rhabdovirus (ChMRV), which was only detected in the transcriptome dataset of the maize leafhopper collected in Argentina (Appendix A). The partial negative-sense single-stranded RNA ChMRV genome (GenBank accession number BK068253) is 12,271 nt in length (Table 1, Figure 5A). ChMRV genome has the five canonical rhabdoviral structural protein genes nucleocapsid (N), phosphoprotein (P), matrix protein (M), glycoprotein (G), and polymerase (L), as is the typical architecture within the family *Rhabdoviridae* [57,58]. Two additional genes, named U1 and U2, were detected between the G and L genes. Thus, the genomic organization of ChMRV is 3′-N-P-M-G-U1-U2-L-5’ (Figure 5A), which is similar to the one reported for a few insect-associated rhabdoviruses, such as the mosquito-associated Xiangyun mono-chu-like virus 11 [57] and the leafhopper-associated viruses Guiyang nephotettix cincticeps rhabdovirus 1 (GuNCRV1) (Feng, unpublished).

The ORF1, which is 1365 nt in length, encodes a putative N protein of 454 aa (Table 1). One nuclear localization signal (NLS) was identified at aa positions 405–440. An NLS was also identified in the N protein encoded by the leafhopper-associated Nephotettix cincticeps negative-stranded RNA virus-1 (NcNSRV1) [32]. The ORF2, which is 936 nt in length, encodes a putative P protein of 311 aa (Table 1). One NLS was identified at aa positions 103–132. An NLS was also identified in the P protein encoded by NcNSRV1 and Spodoptera frugiperda rhabdovirus [32,59]. The ORF3, which is 729 nt in length, encodes a putative M protein of 242 aa (Table 1). One NLS was identified at aa positions 232–241. An NLS was also identified in the M protein encoded by Vesicular stomatitis virus [60]. The ORF4, which is 1989 nt in length, encodes a putative G protein of 662 aa (Table 1). A signal peptide was predicted in the N-terminus of this protein at aa positions 21–22, while a transmembrane domain was predicted in the C-terminus at aa positions 611–633. Signal peptides and transmembrane domains are present in every rhabdovirus glycoproteins, due to its membrane-associated function in the rhabdoviral life cycle [61].

The partial ORF7 encodes the putative L protein (Table 1). Two NLS were identified at aa positions 159–170 and 723–751. An NLS was also identified in the L protein encoded by NcNSRV1 and GuNCRV1 [32]. As stated above, two accessory ORFs were predicted between the G and L genes, which was not surprising, since the G-L gene junction is the most commonly occupied site of accessory genes within rhabdoviruses [62]. Moreover, the presence of two additional genes in both ChMRV and GuNCRV1 could be evidence of gene duplication, which has been mentioned as a mechanism involved in the evolutionary process in other rhabdoviruses [63]. The ORF5, which is 273 nt in length, encodes a putative U1 protein of 90 aa (Table 1). A transmembrane protein was predicted at aa positions 54–76. This protein has the structural characteristics of a class I viroporin, that has been speculated to display membrane-associated functions [64,65]. The ORF6, which is 159 nt in length, encodes a putative U1 protein of 52 aa (Table 1). A transmembrane protein was predicted at aa positions 20–42. Thus, like U1, this protein also should have membrane-associated functions. As expected, transmembrane domains were also identified in the U1 and U2 proteins encoded by GuNCRV1.

BlastP analysis of each protein encoded by the ChMRV genome showed that the highest identities of the N, M, G, and L proteins were with the GuNCRV1 homologous proteins, with low-sequence identities (Table 1). On the other hand, both P and P5 proteins have no hits with any protein present in the NCBI database.

The consensus gene junction 3′-UUAUUUUUUGGUAA-5′ was identified in the non-coding regions of ChMRV genome. This sequence is highly similar to that reported for GuNCRV1 and was used to identify the accessory genes as bona fide predictions.

To explore the relationships of ChMRV with members belonging to the family *Rhabdoviridae*, phylogenetic analysis based on the L protein sequence was carried out, which showed that ChMRV is clustered with the leafhopper-associated GuNCRV1, and the froghopper-associated hemipteran rhabdo-related virus OKIAV30 (Figure 5B). These viruses should be classified into a novel genus within the subfamily *Deltarhabdovirinae* in the family *Rhabdoviridae*. Based on the phylogenetic insights and the observed genetic distance of the newly identified viruses, we tentatively propose an aa sequence identity of 80% in the L protein as threshold for species demarcation in this newly proposed genus.

## 4. Concluding Remarks

Analysis of publicly available RNA-seq data led to the identification of several novel viruses which have expanded the knowledge about the virosphere, shedding light on the diversity and phylogenetic relationships of viruses infecting a wide number of host including insects [10,11,12,13,14,15]. The six viruses associated with the maize leafhopper were related with other leafhopper-associated viruses, thus supporting the host assignments of those viruses identified in this study. Knowledge of the virome (insect viral community) of insect pests is essential to develop bio-insecticides based on viruses; therefore, the identification of the maize leafhopper virome would be useful to develop novel and sustainable approaches based on virocontrol to manage this agronomically important pest.

## Figures and Tables

**Figure 1 viruses-16-01583-f001:**
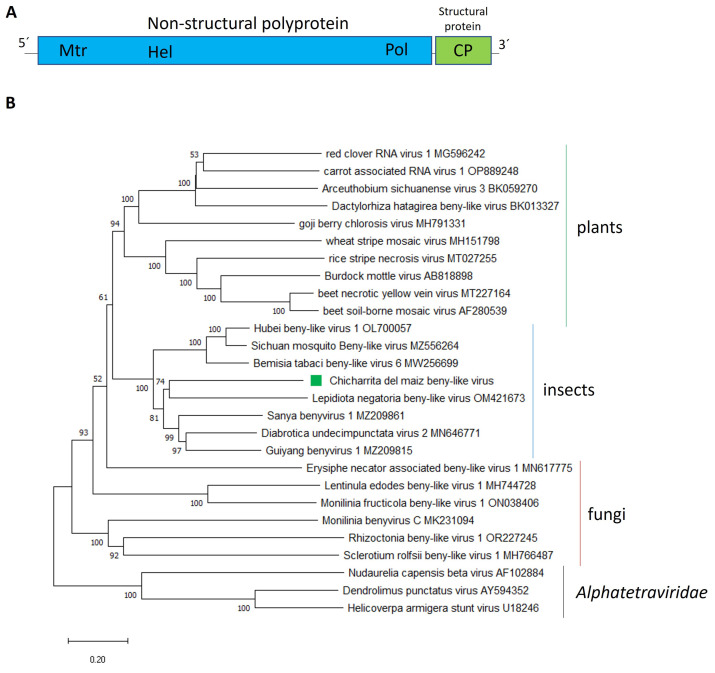
(**A**) Schematic representation of the genome organization of Chicharrita del maiz beny-like virus. Mtr, methyltransferase; Hel, helicase; Pol, RNA polymerase; CP, capsid protein. (**B**) Maximum likelihood phylogenetic trees reconstructed using the RdRp protein sequence of Chicharrita del maiz beny-like virus and of representative beny-like viruses. Bootstrap values above 50% are shown (1000 replicates). Chicharrita del maiz beny-like virus is indicated with a green square. The scale bar shows the substitution per site.

**Figure 2 viruses-16-01583-f002:**
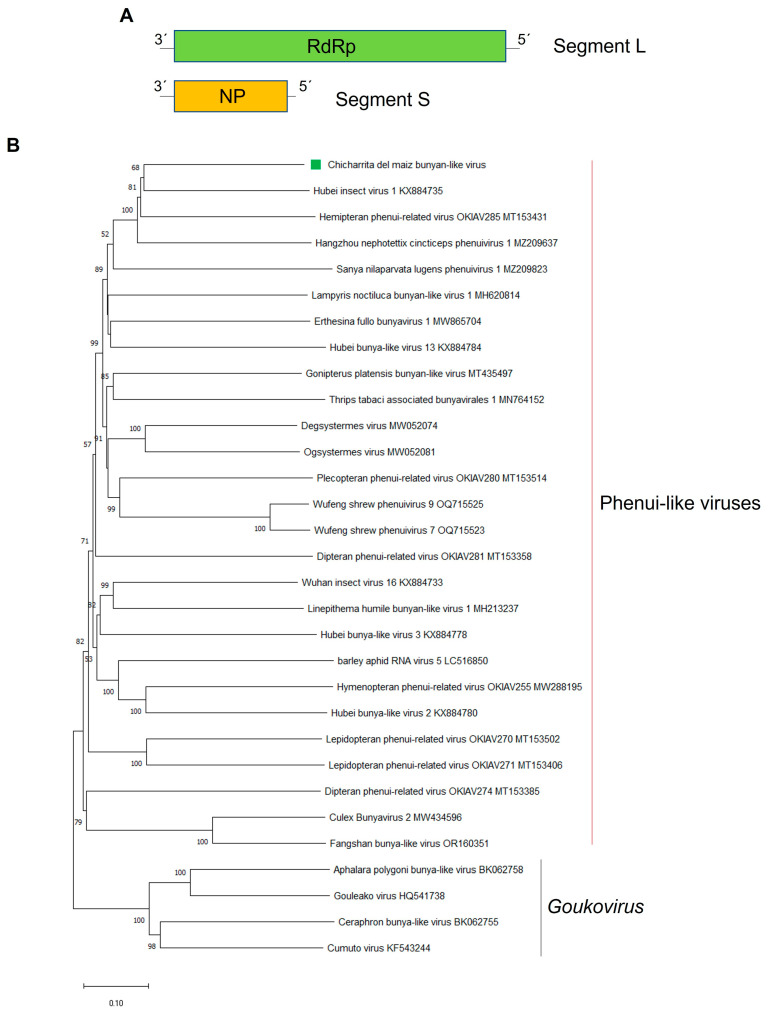
(**A**) Schematic representation of the genome organization of Chicharrita del maiz bunyan-like virus. (**B**) Maximum likelihood phylogenetic trees reconstructed using the RdRp protein sequence of Chicharrita del maiz bunyan-like virus and of representative bunyan-like viruses. Bootstrap values above 50% are shown (1000 replicates). Chicharrita del maiz bunyan-like virus is indicated with a green square. The scale bar shows the substitution per site.

**Figure 3 viruses-16-01583-f003:**
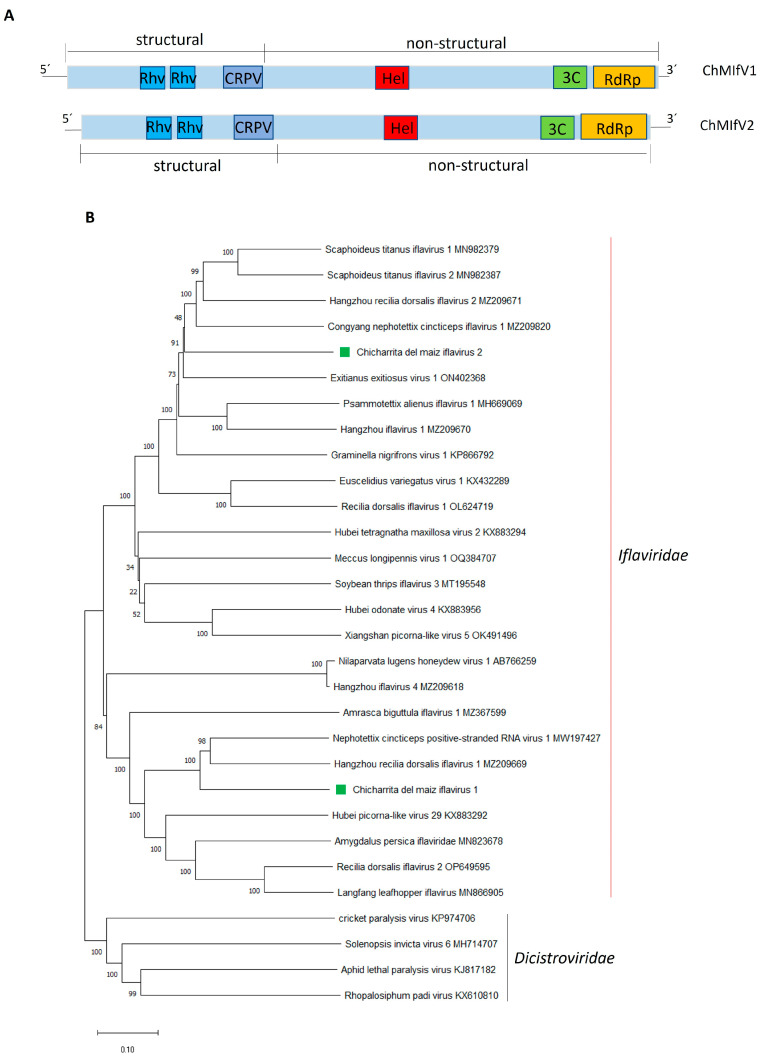
(**A**) Schematic representation of the genome organization of Chicharrita del maiz iflavirus 1 and Chicharrita del maiz iflavirus 2. Rhv, rhinovirus-like capsid protein; CRPV, cricket paralysis virus-like capsid; Hel, helicase; 3C, 3C-like protease; RdRp, RNA-directed RNA polymerase; (**B**) Maximum likelihood phylogenetic trees reconstructed using the RdRp protein sequence of Chicharrita del maiz iflavirus 1 and Chicharrita del maiz iflavirus 2 and of representative iflaviruses. Bootstrap values above 50% are shown (1000 replicates). Chicharrita del maiz iflavirus 1 and Chicharrita del maiz iflavirus 2 are indicated with green squares. The scale bar shows the substitution per site.

**Figure 4 viruses-16-01583-f004:**
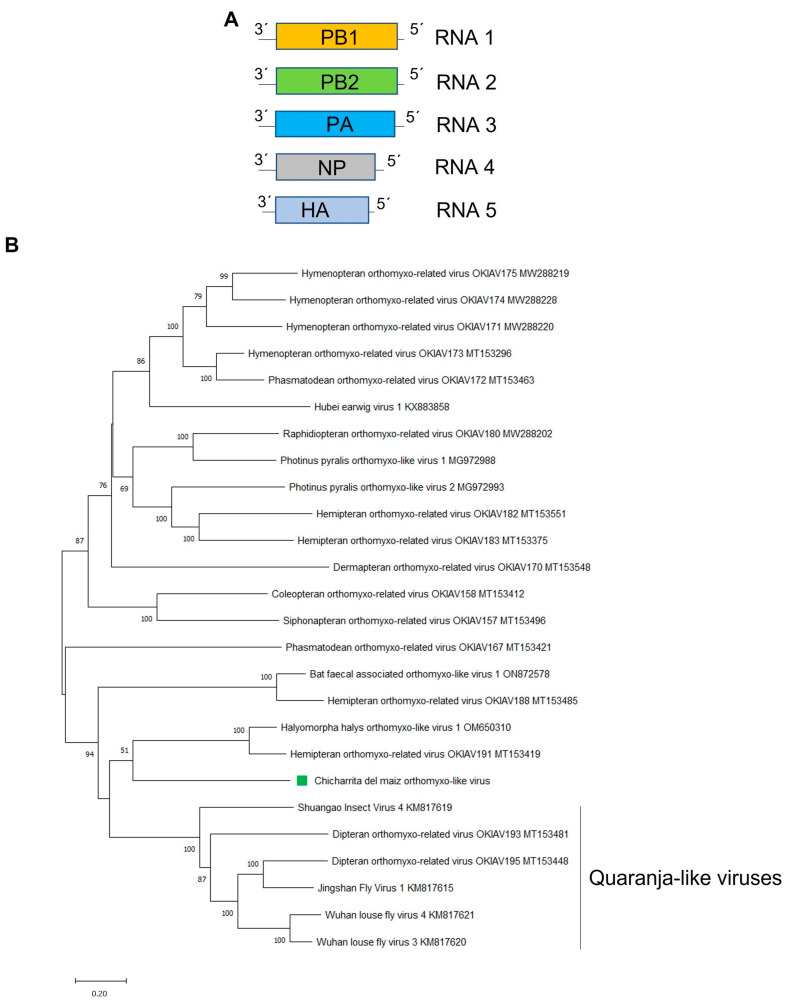
(**A**) Schematic representation of the genome organization of Chicharrita del maiz orthomyxo-like virus. (**B**) Maximum likelihood phylogenetic trees reconstructed using the PB1 proteins sequence of Chicharrita del maiz orthomyxo -like virus and of representative orthomyxo -like viruses. Bootstrap values above 50% are shown (1000 replicates). Chicharrita del maiz orthomyxo -like virus is indicated with a green square. The scale bar shows the substitution per site.

**Figure 5 viruses-16-01583-f005:**
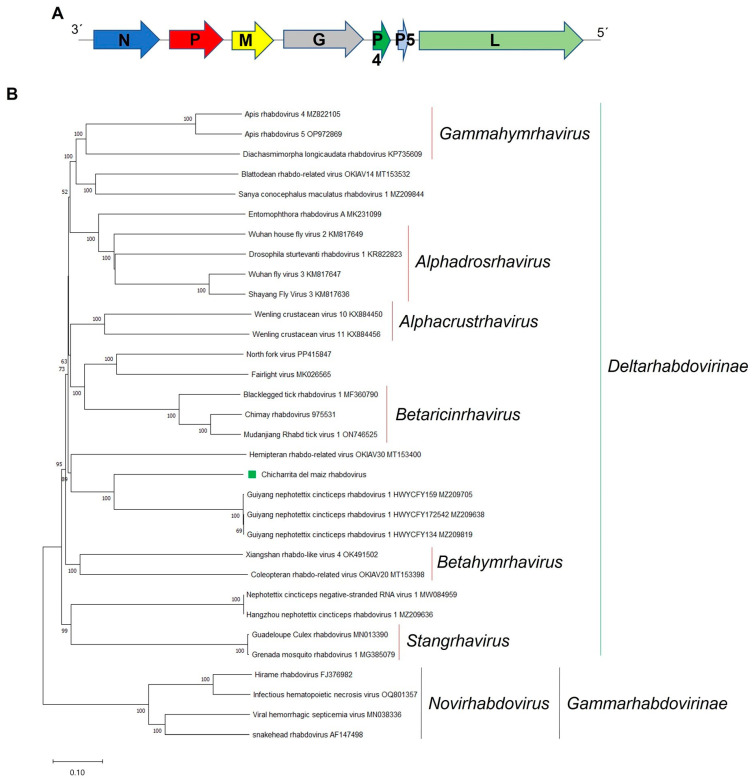
(**A**) Schematic representation of the genome organization of Chicharrita del maiz rhabdovirus. (**B**) Maximum likelihood phylogenetic trees reconstructed using the L protein sequence of Chicharrita del maiz rhabdovirus and of representative rhabdoviruses. Bootstrap values above 50% are shown (1000 replicates). Chicharrita del maiz rhabdovirus is indicated with a green square. The scale bar shows the substitution per site.

**Table 1 viruses-16-01583-t001:** Summary of the *Dalbulus maidis* viruses identified and genomic features.

Virus Name/Abbreviation	Accession Number	Genome/Segment Length (nt)	Protein ID/Length (aa)	Highest Scoring Virus-Protein/*E*-Value/Query Coverage%/Identity % (Blast P)
Chicharrita del maíz beny-like virus/ChMBLV	BK068250	6887	Polyprotein/2003 structural P/224	DuV2-polyprotein/0.0/70/46.61 DuV2-structural P/8 × 10^−7^/53/29.75
Chicharrita del maíz bunyan-like virus/ChMBV	BK068254 BK068255	Segment L 6651 Segment S 1281	RdRp/2166 NP/327	HbIV1-RdRp/0.0/99/49.84 HbIV1-NP/8 × 10^−53^/81/38.01
Chicharrita del maíz iflavirus 1/ChMIfV1	BK068251	11,104	Polyprotein/3306	NcPSRV1-polyprotein/0.0/96/64.03
Chicharrita del maíz iflavirus 2/ChMIfV2	BK068252	10,318	Polyprotein/3102	StIfV1-polyprotein/0.0/99/53.04
Chicharrita del maíz orthomyxo-like virus/ChMOMLV	BK068245 BK068246 BK068247 BK068248 BK068249	RNA1 2557 RNA2 2460 RNA3 2348 RNA4 1817 RNA5 1681	PB1/794 PB2/777 PA/739 NP/572 HA/503	HhOlV1-PB1/0.0/94/49.41 HhOlV1-PB2/4 × 10^−82^/98/26.32 BFaOMLV1-PA/4 × 10^−123^/99/34.59 HmOMRVOKIAV183-NP/3 × 10^−60^/89/29.87 Cotesiavirus orthomyxi-HA/2 × 10^−20^/73/24.56
Chicharrita del maíz rhabdovirus/ChMRV	BK068253	12,271 *	N/454 P/311 M/242 G/662 U1/90 U2/52 L/2082 *	GuNCRV1-N/1 × 10^−59^/92/31.88 no hits GuNCRV1-M/3 × 10^−28^/74/29.83 GuNCRV1-G/7 × 10^−144^/95/38.79 no hits no hits GuNCRV1-L/0.0/99/48.44

Abbreviations are found within the main text. * partial sequence.

## Data Availability

The sequences deposited in GenBank with accession numbers BK068245-BK068255 have been uploaded as Appendix A of this work.

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
