# Peer review of "Insights into the RNA Virome of the Corn Leafhopper Dalbulus maidis, a Major Emergent Threat of Maize in Latin America"

_viruses, 2024, doi:10.3390/v16101583_

Round 1

Reviewer 1 Report

Comments and Suggestions for Authors

This paper describes the putative novel RNA viruses of the corn leafhopper (Dalbulus maidis). The authors used the sequence archives available in a public database to assemble and analyze the viral genomes. The manuscript contains essential new information and may be published after a minor revision.

L 154-162 and Fig. 1, Beny-like viruses – Benyvirus replicases typically contain methyltransferase and papain-like cysteine proteinase domains. Did the authors look for these domains in the Beny-like sequences they described? If the Mtr and Pro were not found, this must be discussed. In Fig. 1, the conserved domains of Mtr, Hel, Pol, and Pro (if there are Mtr and/or Pro) must be indicated.

L 205-208 – The newly assembled bunya-like virus is most closely related to bunyavirids with bipartite genomes, thus indicating that it is also bipartite. Is there a chance that a third genomic component coding for glycoprotein(s) escaped the sequence analysis? This may be tricky when the genome structure of a multipartite RNA virus is deduced solely from deep sequencing data without classical tests, such as the virus isolation and RNA analysis. A discussion of this matter is in order here. The same applies to the description of the orthomyxo-like virus in this manuscript.

Fig. 3 – The abbreviations for the protein domains must be deciphered in the Figure legend.

Author Response

This paper describes the putative novel RNA viruses of the corn leafhopper (Dalbulus maidis). The authors used the sequence archives available in a public database to assemble and analyze the viral genomes. The manuscript contains essential new information and may be published after a minor revision.

We thank reviewer #1 for taking the time to thoroughly assess our MS and provide suggestions which improved the MS.

1-     L 154-162 and Fig. 1, Beny-like viruses – Benyvirus replicases typically contain methyltransferase and papain-like cysteine proteinase domains. Did the authors look for these domains in the Beny-like sequences they described? If the Mtr and Pro were not found, this must be discussed. In Fig. 1, the conserved domains of Mtr, Hel, Pol, and Pro (if there are Mtr and/or Pro) must be indicated.

The MT domain was identified, and the corresponding text was included in the revised version of the manuscript. Fig.1 was modified accordingly to indicate the conserved domains found. The papain-like cysteine protease domain was neither identified in the dalbulus benyvirus nor in related viruses; thus a sentence that reads as “Unlike its plant-associated counterparts [37, 41], the insect-associated beny-like virus replicase, like those encoded by ChMBLV and its related viruses [38-40, 42], do not have any papain-like protease domain in their sequences” was added to the manuscript.  

2-     The newly assembled bunya-like virus is most closely related to bunyavirids with bipartite genomes, thus indicating that it is also bipartite. Is there a chance that a third genomic component coding for glycoprotein(s) escaped the sequence analysis? This may be tricky when the genome structure of a multipartite RNA virus is deduced solely from deep sequencing data without classical tests, such as the virus isolation and RNA analysis. A discussion of this matter is in order here. The same applies to the description of the orthomyxo-like virus in this manuscript.

We indicated that the novel bunya-like virus is bipartite. In Lines 209-213 there is a sentence that reads “Bunyavirids have different number of genome segments; however, those viruses phylogenetically related with ChMBV, are all bisegmented RNA viruses [11]. Thus, these leafhopper-associated bunya-like viruses appear to be a new cluster of bis-egmented RNA viruses that could be classified into a new genus within the Bunyavirales order.”.

As indicated in the sentence above, any third segment was neither identified in those viruses related with the dalbulus bunyan-like. A sentence, that reads “Intriguingly, the segment coding for the glycoprotein was not identified in ChMBV and in those related viruses”.  

Regarding the orthomyxo-like virus, related viruses have also 5 segments, as was stated in the manuscript. Thus, we do not consider necessary to add any novel sentence.

Fig. 3 – The abbreviations for the protein domains must be deciphered in the Figure legend.

We added the abbreviations of the proteins domain in the Figure 3 legend.

Reviewer 2 Report

Comments and Suggestions for Authors

This manuscript is well written, and the newly identified viruses are well characterized. In a perfect world, it would be nice to have some biological data but given the nature of the paper, the authors have done a good job. Since the authors are characterizing novel viruses and assigning new names, the one thing I see missing is defining the ICTV criteria. ICTV has outlined criteria for species demarcation based on the percent identity at the gene or polyprotein level.

For each novel virus described in this manuscript, I would like to see how they qualify as a novel virus, which ICTV criteria they fulfill, and also provide the percent identities. For example, the species demarcation criteria for members of benyviruses are less than 90% CP protein identity. I do not doubt that the new beny-like virus the authors described meets this criterion, it would make the manuscript better if those criteria were spelled out and how the novel viruses fall within those criteria are mentioned.

Author Response

This manuscript is well written, and the newly identified viruses are well characterized. In a perfect world, it would be nice to have some biological data but given the nature of the paper, the authors have done a good job.

We thank reviewer #2 for taking the time to thoroughly assess our MS and provide suggestions which improved the MS.

Since the authors are characterizing novel viruses and assigning new names, the one thing I see missing is defining the ICTV criteria. ICTV has outlined criteria for species demarcation based on the percent identity at the gene or polyprotein level. For each novel virus described in this manuscript, I would like to see how they qualify as a novel virus, which ICTV criteria they fulfill, and also provide the percent identities.

For those viruses that we are suggesting the creation of new genera to accommodate viruses (beny-like, bunya-like, orthomyxo-like and rhabdo), we added a sentence  proposing the demarcation criteria; while for those viruses  that belong to established genera (iflaviruses) we added the sentence indicating that viruses are new species according to the criteria used for species demarcation and we also proposed how those species should be named.